# Variance Reduction for Policy Gradient with Action-Dependent Factorized Baselines

**Cathy Wu**[1], **Aravind Rajeswaran**[2], **Yan Duan**[13], **Vikash Kumar**[2],
**Alexandre M Bayen**[14], **Sham Kakade**[2], **Igor Mordatch**[3], **Pieter Abbeel**[13]
cathywu@eecs.berkeley.edu, aravraj@cs.washington.edu,
rockyduan@eecs.berkeley.edu, vikash@cs.washington.edu,
bayen@berkeley.edu, sham@cs.washington.edu,
igor.mordatch@gmail.com, pabbeel@cs.berkeley.edu
[1] Department of EECS, UC Berkeley
[2] Department of CSE, University of Washington
[3] OpenAI
[4] Institute for Transportation Studies, UC Berkeley

## Abstract

Policy gradient methods have enjoyed great success in deep reinforcement learning but suffer from high variance of gradient estimates. The high variance problem is particularly exasperated in problems with long horizons or high-dimensional action spaces. To mitigate this issue, we derive a bias-free action-dependent baseline for variance reduction which fully exploits the structural form of the stochastic policy itself and does not make any additional assumptions about the MDP. We demonstrate and quantify the benefit of the action-dependent baseline through both theoretical analysis as well as numerical results, including an analysis of the suboptimality of the optimal state-dependent baseline. The result is a computationally efficient policy gradient algorithm, which scales to high-dimensional control problems, as demonstrated by a synthetic 2000-dimensional target matching task. Our experimental results indicate that action-dependent baselines allow for faster learning on standard reinforcement learning benchmarks and high-dimensional hand manipulation and synthetic tasks. Finally, we show that the general idea of including additional information in baselines for improved variance reduction can be extended to partially observed and multi-agent tasks.

## 1 Introduction

Deep reinforcement learning has achieved impressive results in recent years in domains such as video games from raw visual inputs (Mnih et al., 2015), board games (Silver et al., 2016), simulated control tasks (Schulman et al., 2016; Lillicrap et al., 2016; Rajeswaran et al., 2017a), and robotics (Levine et al., 2016). An important class of methods behind many of these success stories are policy gradient methods (Williams, 1992; Sutton et al., 2000; Kakade, 2002; Schulman et al., 2015; Mnih et al., 2016), which directly optimize parameters of a stochastic policy through local gradient information obtained by interacting with the environment using the current policy. Policy gradient methods operate by increasing the log probability of actions proportional to the future rewards influenced by these actions. On average, actions which perform better will acquire higher probability, and the policy's expected performance improves.

A critical challenge of policy gradient methods is the high variance of the gradient estimator. This high variance is caused in part due to difficulty in credit assignment to the actions which affected the future rewards. Such issues are further exacerbated in long horizon problems, where assigning credits properly becomes even more challenging. To reduce variance, a "baseline" is often employed, which allows us to increase or decrease the log probability of actions based on whether they perform better or worse than the average performance when starting from the same state. This is particularly useful in long horizon problems, since the baseline helps with temporal credit assignment by

removing the influence of future actions from the total reward. A better baseline, which predicts the average performance more accurately, will lead to lower variance of the gradient estimator.

The key insight of this paper is that when the individual actions produced by the policy can be decomposed into multiple factors, we can incorporate this additional information into the baseline to further reduce variance. In particular, when these factors are conditionally independent given the current state, we can compute a separate baseline for each factor, whose value can depend on all quantities of interest except that factor. This serves to further help credit assignment by removing the influence of other factors on the rewards, thereby reducing variance. In other words, information about the other factors can provide a better evaluation of how well a specific factor performs. Such factorized policies are very common, with some examples listed below.

- In continuous control and robotics tasks, multivariate Gaussian policies with a diagonal covariance matrix are often used. In such cases, each action coordinate can be considered a factor. Similarly, factorized categorical policies are used in game domains like board games and Atari.
- In multi-agent and distributed systems, each agent deploys its own policy, and thus the actions of each agent can be considered a factor of the union of all actions (by all agents). This is particularly useful in the recent emerging paradigm of centralized learning and decentralized execution (Foerster et al., 2017; Lowe et al., 2017). In contrast to the previous example, where factorized policies are a common design choice, in these problems they are dictated by the problem setting.

We demonstrate that action-dependent baselines consistently improve the performance compared to baselines that use only state information. The relative performance gain is task-specific, but in certain tasks, we observe significant speed-up in the learning process. We evaluate our proposed method on standard benchmark continuous control tasks, as well as on a high-dimensional door opening task with a five-fingered hand, a synthetic high-dimensional target matching task, on a blind peg insertion POMDP task, and a multi-agent communication task. We believe that our method will facilitate further applications of reinforcement learning methods in domains with extremely high-dimensional actions, including multi-agent systems. Videos and additional results of the paper are available at `https://sites.google.com/view/ad-baselines`.

## 2  RELATED WORKS

Three main classes of methods for reinforcement learning include value-based methods (Watkins & Dayan, 1992), policy-based methods (Williams, 1992; Kakade, 2002; Schulman et al., 2015), and actor-critic methods (Konda & Tsitsiklis, 2000; Peters & Schaal, 2008; Mnih et al., 2016). Value-based and actor-critic methods usually compute a gradient of the objective through the use of critics, which are often biased, unless strict compatibility conditions are met (Sutton et al., 2000; Konda & Tsitsiklis, 2000). Such conditions are rarely satisfied in practice due to the use of stochastic gradient methods and powerful function approximators. In comparison, policy gradient methods are able to compute an unbiased gradient, but suffer from high variance. Policy gradient methods are therefore usually less sample efficient, but can be more stable than critic-based methods (Duan et al., 2016).

A large body of work has investigated variance reduction techniques for policy gradient methods. One effective method to reduce variance without introducing bias is through using a baseline, which has been widely studied (Sutton & Barto, 1998; Weaver & Tao, 2001; Greensmith et al., 2004; Schulman et al., 2016). However, fully exploiting the factorizability of the policy probability distribution to further reduce variance has not been studied. Recently, methods like Q-Prop (Gu et al., 2017) make use of an action-dependent control variate, a technique commonly used in Monte Carlo methods and recently adopted for RL. Since Q-Prop utilizes off-policy data, it has the potential to be more sample efficient than pure on-policy methods. However, Q-prop is significantly more computationally expensive, since it needs to perform a large number of gradient updates on the critic using the off-policy data, thus not suitable with fast simulators. In contrast, our formulation of action-dependent baselines has little computational overhead, and improves the sample efficiency compared to on-policy methods with state-only baseline.

The idea of using additional information in the baseline or critic has also been studied in other contexts. Methods such as Guided Policy Search (Levine et al., 2016; Mordatch et al., 2015) and variants train policies that act on high-dimensional observations like images, but use a low dimensional encoding of the problem like joint positions during the training process. Recent efforts in multi-agent

systems (Foerster et al., 2017; Lowe et al., 2017) also use additional information in the centralized training phase to speed-up learning. However, using the structure in the policy parameterization itself to enhance the learning speed, as we do in this work, has not been explored.

## 3 PRELIMINARIES

In this section, we establish the notations used throughout this paper, as well as basic results for policy gradient methods, and variance reduction via baselines.

### 3.1 NOTATION

This paper assumes a discrete-time Markov decision process (MDP), defined by $(\mathcal{S}, \mathcal{A}, \mathcal{P}, r, \rho_0, \gamma)$, in which $\mathcal{S} \subseteq \mathbb{R}^n$ is an $n$-dimensional state space, $\mathcal{A} \subseteq \mathbb{R}^m$ an $m$-dimensional action space, $\mathcal{P} : \mathcal{S} \times \mathcal{A} \times \mathcal{S} \to \mathbb{R}_+$ a transition probability function, $r : \mathcal{S} \times \mathcal{A} \to \mathbb{R}$ a bounded reward function, $\rho_0 : \mathcal{S} \to \mathbb{R}_+$ an initial state distribution, and $\gamma \in (0, 1]$ a discount factor. The presented models are based on the optimization of a stochastic policy $\pi_\theta : \mathcal{S} \times \mathcal{A} \to \mathbb{R}_+$ parameterized by $\theta$. Let $\eta(\pi_\theta)$ denote its expected return: $\eta(\pi_\theta) = \mathbb{E}_\tau[\sum_{t=0}^\infty \gamma^t r(s_t, a_t)]$, where $\tau = (s_0, a_0, \dots)$ denotes the whole trajectory, $s_0 \sim \rho_0(s_0)$, $a_t \sim \pi_\theta(a_t|s_t)$, and $s_{t+1} \sim \mathcal{P}(s_{t+1}|s_t, a_t)$ for all $t$. Our goal is to find the optimal policy $\arg\max_\theta \eta(\pi_\theta)$. We will use $\hat{Q}(s_t, a_t)$ to describe samples of cumulative discounted return, and $Q(a_t, s_t)$ to describe a function approximation of $\hat{Q}(s_t, a_t)$. We will use "Q-function" when describing an abstract action-value function.

For a partially observable Markov decision process (POMDP), two more components are required, namely $\Omega$, a set of observations, and $\mathcal{O} : \mathcal{S} \times \Omega \to \mathbb{R}_{\geq 0}$, the observation probability distribution. In the fully observable case, $\Omega \equiv \mathcal{S}$. Though the analysis in this article is written for policies over states, the same analysis can be done for policies over observations.

### 3.2 THE SCORE FUNCTION (SF) ESTIMATOR

An important technique used in the derivation of the policy gradient is known as the score function (SF) estimator (Williams, 1992), which also comes up in the justification of baselines. Suppose that we want to estimate $\nabla_\theta \mathbb{E}_x[f(x)]$ where $x \sim p_\theta(x)$, and the family of distributions $\{p_\theta(x) : \theta \in \Theta\}$ has common support. Further suppose that $\log p_\theta(x)$ is continuous in $\theta$. In this case we have

$$\nabla_\theta \mathbb{E}_x[f(x)] = \nabla_\theta \int p_\theta(x)f(x)dx = \int p_\theta(x)\frac{\nabla_\theta p_\theta(x)}{p_\theta(x)}f(x)dx$$

$$= \int p_\theta(x)\nabla_\theta \log p_\theta(x)f(x)dx = \mathbb{E}_x\left[\nabla_\theta \log p_\theta(x)f(x)\right]. \tag{1}$$

### 3.3 POLICY GRADIENT

The Policy Gradient Theorem (Sutton et al., 2000) states that

$$\nabla_\theta \eta(\pi_\theta) = \mathbb{E}_\tau \left[ \sum_{t=0}^\infty \nabla_\theta \log \pi_\theta(a_t|s_t) \sum_{t'=t}^\infty \gamma^{t'-t} r_{t'} \right]. \tag{2}$$

For convenience, define $\rho_\pi(s) = \sum_{t=0}^\infty \gamma^t p(s_t = s)$ as the state visitation frequency, and $\hat{Q}(s_t, a_t) = \sum_{t'=t}^\infty \gamma^{t'-t} r_{t'}$. We can rewrite the above equation (with abuse of notation) as

$$\nabla_\theta \eta(\pi_\theta) = \mathbb{E}_{\rho_\pi, \pi} \left[ \nabla_\theta \log \pi_\theta(a_t|s_t)\hat{Q}(s_t, a_t) \right]. \tag{3}$$

It is further shown that we can reduce the variance of this gradient estimator without introducing bias by subtracting off a quantity dependent on $s_t$ from $\hat{Q}(s_t, a_t)$ (Williams, 1992; Greensmith et al., 2004). See Appendix A for a derivation of the optimal state-dependent baseline.

$$\nabla_\theta \eta(\pi_\theta) = \mathbb{E}_{\rho_\pi, \pi} \left[ \nabla_\theta \log \pi_\theta(a_t|s_t) \left( \hat{Q}(s_t, a_t) - b(s_t) \right) \right] \tag{4}$$

This is valid because, applying the SF estimator in the opposite direction, we have

$$\mathbb{E}_{a_t} \left[ \nabla_\theta \log \pi_\theta(a_t|s_t)b(s_t) \right] = \nabla_\theta \mathbb{E}_{a_t} \left[ b(s_t) \right] = 0 \tag{5}$$

## 4  ACTION-DEPENDENT BASELINES

In practice there can be rich internal structure in the policy parameterization. For example, for continuous control tasks, a very common parameterization is to make $\pi_\theta(a_t|s_t)$ a multivariate Gaussian with diagonal variance, in which case each dimension $a_t^i$ of the action $a_t$ is conditionally independent of other dimensions, given the current state $s_t$. Another example is when the policy outputs a tuple of discrete actions with factorized categorical distributions. In the following subsections, we show that such structure can be exploited to further reduce the variance of the gradient estimator without introducing bias by changing the form of the baseline. Then, we derive the optimal action-dependent baseline for a class of problems and analyze the suboptimality of non-optimal baselines in terms of variance reduction. We then propose several practical baselines for implementation purposes. We conclude the section with the overall policy gradient algorithm with action-dependent baselines for factorized policies. We provide an exposition for situations when the conditional independence assumption does not hold, such as for stochastic policies with general covariance structures, in Appendix E, and for compatibility with other variance reduction techniques in Appendix F.

### 4.1  BASELINES FOR POLICIES WITH CONDITIONALLY INDEPENDENT FACTORS

In the following, we analyze action-dependent baselines for policies with conditionally independent factors. For example, multivariate Gaussian policies with a diagonal covariance structure are commonly used in continuous control tasks. Assuming an $m$-dimensional action space, we have $\pi_\theta(a_t|s_t) = \prod_{i=1}^m \pi_\theta(a_t^i|s_t)$. Hence

$$\nabla_\theta \eta(\pi_\theta) = \mathbb{E}_{\rho_\pi,\pi}\left[\nabla_\theta \log \pi_\theta(a_t|s_t)\hat{Q}(s_t,a_t)\right] = \mathbb{E}_{\rho_\pi,\pi}\left[\sum_{i=1}^m \nabla_\theta \log \pi_\theta(a_t^i|s_t)\hat{Q}(s_t,a_t)\right] \quad (6)$$

In this case, we can set $b_i$, the baseline for the $i$th factor, to depend on all other actions in addition to the state. Let $a_t^{-i}$ denote all dimensions other than $i$ in $a_t$ and denote the $i$th baseline by $b_i(s_t, a_t^{-i})$. Due to conditional independence and the score function estimator, we have

$$\mathbb{E}_{a_t}\left[\nabla_\theta \log \pi_\theta(a_t^i|s_t)b_i(s_t,a_t^{-i})\right] = \mathbb{E}_{a_t^{-i}}\left[\nabla_\theta \mathbb{E}_{a_t^i}\left[b_i(s_t,a_t^{-i})\right]\right] = 0 \quad (7)$$

Hence we can use the following gradient estimator

$$\nabla_\theta \eta(\pi_\theta) = \mathbb{E}_{\rho_\pi,\pi}\left[\sum_{i=1}^m \nabla_\theta \log \pi_\theta(a_t^i|s_t)\left(\hat{Q}(s_t,a_t) - b_i(s_t,a_t^{-i})\right)\right] \quad (8)$$

This is compatible with advantage function form of the policy gradient (Schulman et al., 2016):

$$\nabla_\theta \eta(\pi_\theta) = \mathbb{E}_{\rho_\pi,\pi}\left[\sum_{i=1}^m \nabla_\theta \log \pi_\theta(a_t^i|s_t)\hat{A}_i(s_t,a_t)\right] \quad (9)$$

where $\hat{A}_i(s_t,a_t) = Q(s_t,a_t) - b_i(s_t,a_t^{-i})$. Note that the policy gradient now comprises of $m$ component policy gradient terms, each with a different advantage term.

In Appendix E, we show that the methodology also applies to general policy structures (for example, a Gaussian policy with a general covariance structure), where the conditional independence assumption does not hold. The result is bias-free albeit different baselines.

### 4.2  OPTIMAL ACTION-DEPENDENT BASELINE

In this section, we derive the optimal action-dependent baseline and show that it is better than the state-only baseline. We seek the optimal baseline to minimize the variance of the policy gradient estimate. First, we write out the variance of the policy gradient under any action-dependent baseline. Let us define $z_i := \nabla_\theta \log \pi_\theta(a_t^i|s_t)$ and the component policy gradient:

$$\nabla \eta_i(\pi_\theta) := \mathbb{E}_{\rho_\pi,\pi}\left[\nabla_\theta \log \pi_\theta(a_t^i|s_t)\left(\hat{Q}(s_t,a_t) - b_i(s_t,a_t^{-i})\right)\right]. \quad (10)$$

For simplicity of exposition, we make the following assumption:

$$\nabla_\theta \log \pi_\theta(a_t^i|s_t)^T \nabla_\theta \log \pi_\theta(a_t^j|s_t) \equiv z_i^T z_j = 0, \quad \forall i \neq j \quad (11)$$

which translates to meaning that different subsets of parameters strongly influence different action dimensions or factors. We note that this assumption is primarily for the theoretical analysis to be clean, and is not required to run the algorithm in practice. In particular, even without this assumption, the proposed baseline is bias-free. When the assumption holds, the optimal action-dependent baseline can be analyzed thoroughly. Some examples where these assumptions do hold include multi-agent settings where the policies are conditionally independent by construction, cases where the policy acts based on independent components (Cao et al., 2007) of the observation space, and cases where different function approximators are used to control different actions or synergies (Todorov & Ghahramani, 2004; Todorov et al., 2005) without weight sharing.

The optimal action-dependent baseline is then derived to be:

$$b_i^*(s_t, a_t^{-i}) = \frac{\mathbb{E}_{a_t^i}\left[\nabla_\theta \log \pi_\theta(a_t^i|s_t)^T \nabla_\theta \log \pi_\theta(a_t^i|s_t)\hat{Q}(s_t, a_t)\right]}{\mathbb{E}_{a_t^i}\left[\nabla_\theta \log \pi_\theta(a_t^i|s_t)^T \nabla_\theta \log \pi_\theta(a_t^i|s_t)\right]}. \tag{12}$$

See Appendix B for the full derivation. Since the optimal action-dependent baseline is different for different action coordinates, it is outside the family of state-dependent baselines barring pathological cases.

## 4.3 SUBOPTIMALITY OF THE OPTIMAL STATE-DEPENDENT BASELINE

How much do we reduce variance over a traditional baseline that only depends on state? We use the following notation:

$$Z_i := Z_i(s_t, a_t^{-i}) = \mathbb{E}_{a_t^i}\left[\nabla_\theta \log \pi_\theta(a_t^i|s_t)^T \nabla_\theta \log \pi_\theta(a_t^i|s_t)\right] \tag{13}$$

$$Y_i := Y_i(s_t, a_t^{-i}) = \mathbb{E}_{a_t^i}\left[\nabla_\theta \log \pi_\theta(a_t^i|s_t)^T \nabla_\theta \log \pi_\theta(a_t^i|s_t)\hat{Q}(s_t, a_t)\right] \tag{14}$$

Then, using Equation (51) (Appendix C), we show the following improvement with the optimal action-dependent baseline:

$$I_{b=b^*(s)} = \sum_i \mathbb{E}_{\rho_\pi, a_t^{-i}}\left[\frac{1}{Z_i}\left(\frac{Z_i}{\sum_j Z_j}\sum_j Y_j - Y_i\right)^2\right] \tag{15}$$

See Appendices C and D for the full derivation. We conclude that the optimal action-dependent baseline does not degenerate into the optimal state-dependent baseline. Equation (15) states that the variance difference is a weighted sum of the deviation of the per-component score-weighted marginalized Q (denoted $Y_i$) from the component weight (based on score only, not Q) of the overall aggregated marginalized Q values (denoted $\sum_j Y_j$). This suggests that the difference is particularly large when the Q function is highly sensitive to the actions, especially along those directions that influence the gradient the most. Our empirical results in Section 5 additionally demonstrate the benefit of action-dependent over state-only baselines.

## 4.4 MARGINALIZATION OF THE GLOBAL ACTION-VALUE FUNCTION

Using the previous theory, we now consider various baselines that could be used in practice and their associated computational cost.

**Marginalized Q baseline** Even though the optimal state-only baseline is known, it is rarely used in practice (Duan et al., 2016). Rather, for both computational and conceptual benefit, the choice of $b(s_t) = \mathbb{E}_{a_t}[\hat{Q}(s_t, a_t)] = V(s_t)$ is often used. Similarly, we propose to use $b_i(s_t, a_t^{-i}) = \mathbb{E}_{a_t^i}\left[\hat{Q}(s_t, a_t)\right]$ which is the action-dependent analogue. In particular, when log probability of each policy factor is loosely correlated with the action-value function, then the proposed baseline is close to the optimal baseline.

$$I_{b=\mathbb{E}_{a_t^i}[\hat{Q}(a_t, s_t)]} = \sum_i \mathbb{E}_{\rho_\pi, a_t^{-i}}\left[Z_i\left(\mathbb{E}_{a^i}\left[\hat{Q}(a_t, s_t)\right] - \frac{\mathbb{E}_{a_t^i}\left[z_i^T z_i\hat{Q}(s_t, a_t)\right]}{\mathbb{E}_{a_t^i}\left[z_i^T z_i\right]}\right)^2\right] \approx 0 \tag{16}$$

when $\mathbb{E}_{a_t^i}\left[z_i^T z_i \hat{Q}(s_t, a_t)\right] \approx \mathbb{E}_{a_t^i}\left[z_i^T z_i\right] \mathbb{E}_{a_t^i}\left[\hat{Q}(s_t, a_t)\right]$.

This has the added benefit of requiring learning only one function approximator, for estimating $Q(s_t, a_t)$, and implicitly using it to obtain the baselines for each action coordinate. That is, $Q(s_t, a_t)$ is a function approximating samples $\hat{Q}(s_t, a_t)$.

**Monte Carlo marginalized Q baseline**  After fitting $Q_{\pi_\theta}(s_t, a_t)$ we can obtain the baselines through Monte Carlo estimates:

$$b_i(s_t, a_t^{-i}) = \frac{1}{M} \sum_{j=0}^{M} Q_{\pi_\theta}(s_t, (a_t^{-i}, \alpha_j)) \tag{17}$$

where $\alpha_j \sim \pi_\theta(a_t^i | s_t)$ are samples of the action coordinate $i$. In general, any function may be used to aggregate the samples, so long as it does not depend on the sample value $a_t^i$. For instance, for discrete action dimensions, the sample max can be computed instead of the mean.

**Mean marginalized Q baseline**  Though we reduced the computational burden from learning $m$ functions to one function, the use of Monte Carlo samples can still be computationally expensive. In particular, when using deep neural networks to approximate the Q-function, forward propagation through the network can be even more computationally expensive than stepping through a fast simulator (e.g. MuJoCo). In such settings, we further propose the following more computationally practical baseline:

$$b_i(s_t, a_t^{-i}) = Q_{\pi_\theta}(s_t, (a_t^{-i}, \bar{a}_t^i)) \tag{18}$$

where $\bar{a}_t^i = \mathbb{E}_{\pi_\theta}\left[a_t^i\right]$ is the average action for coordinate $i$.

### 4.5 FINAL ALGORITHM

The final practical algorithm for fully factorized policies is as follows.

---
**Algorithm 1** Policy gradient for factorized policies using action-dependent baselines

---
**Require:** number of iterations $N$, batch size $B$, initial policy parameters $\theta$
    Initialize action-value function estimate $Q_{\pi_\theta}(s_t, a_t) \equiv 0$ and policy $\pi_\theta$
    **for** $j$ in $\{1, \ldots, N\}$ **do**
        Collect samples: $(s_t, a_t)_{t \in \{1, \ldots, B\}}$
        Compute baseline: $b_i(s_t, a_t^{-i}) = \mathbb{E}_{a_t^i}\left[\hat{Q}(s_t, a_t)\right]$ for $i \in \{1, \ldots, m\}$ [e.g. Equations (17- 18)]
        Compute advantages: $\hat{A}_i(s_t, a_t) := \hat{Q}(s_t, a_t) - b_i(s_t, a_t^{-i}), \forall t$
        Perform a policy update step on $\theta$ using $\hat{A}_i(s_t, a_t)$ [Equation (9)]
        Update action-value function approximation with current batch: $Q_{\pi_\theta}(s_t, a_t)$
    **end for**

---

Computing the baseline can be done with either proposed technique in Section 4.4. A similar algorithm can be written for general policies (Appendix E), which makes no assumptions on the conditional independence across action dimensions.

## 5 EXPERIMENTS AND RESULTS

**Continuous control benchmarks**  Firstly, we present the results of the proposed action-dependent baselines on popular benchmark tasks. These tasks have been widely studied in the deep reinforcement learning community (Duan et al., 2016; Gu et al., 2017; Lillicrap et al., 2016; Rajeswaran et al., 2017b). The studied tasks include the hopper, half-cheetah, and ant locomotion tasks simulated in MuJoCo (Todorov et al., 2012).[1] In addition to these tasks, we also consider a door opening task

---
    [1]We used physics parameters as recommended in Rajeswaran et al. (2017b) and use the MuJoCo 1.5 simulator. Thus the reward numbers may not be consistent with numbers previously reported in literature.

with a high-dimensional multi-fingered hand, introduced in Rajeswaran et al. (2017a) to study the effectiveness of the proposed approach in high-dimensional tasks. Figure 1 presents the learning curves on these tasks. We compare the action-dependent baseline with a baseline that uses only information about the states, which is the most common approach in the literature. We observe that the action-dependent baselines perform consistently better.

A popular baseline parameterization choice is a linear function on a small number of non-linear features of the state (Duan et al., 2016), especially for policy gradient methods. In this work, to enable a fair comparison, we use a Random Fourier Feature representation for the baseline (Rahimi & Recht, 2007; Rajeswaran et al., 2017b). The features are constructed as: $y(x) = \sin(\frac{1}{\nu}Px + \phi)$ where $P$ is a matrix with each element independently drawn from the standard normal distribution, $\phi$ is a random phase shift in $[-\pi, \pi)$ and, and $\nu$ is a bandwidth parameter. These features approximate the RKHS features under an RBF kernel. Using these features, the baseline is parameterized as $b = w^T y(x)$ where $x$ are the appropriate inputs to the baseline, and $w$ are trainable parameters. $P$ and $\phi$ are not trained in this parameterization. Such a representation was chosen for two reasons: (a) we wish to have the same number of trainable parameters for all the baseline architectures, and not have more parameters in the action-dependent case (which has a larger number of inputs to the baseline); (b) since the final representation is linear, it is possible to accurately estimate the optimal parameters with a Newton step, thereby alleviating the results from confounding optimization issues. For policy optimization, we use a variant of the natural policy gradient method as described in Rajeswaran et al. (2017b). See Appendix G for further experimental details.

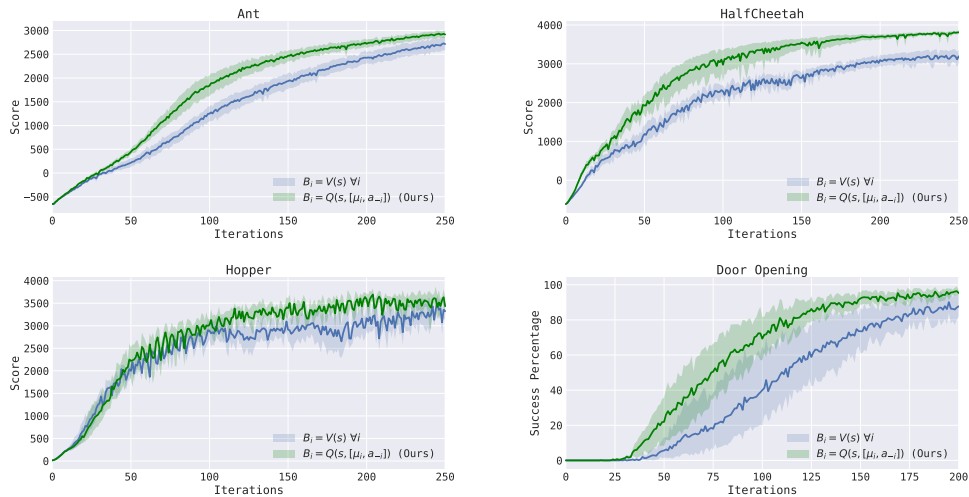

Figure 1: Comparison between value function baseline and action-conditioned baseline on various continuous control tasks. Action-dependent baseline performs consistently better across all the tasks.

**Choice of action-dependent baseline form**    Next, we study the influence of computing the baseline by using empirical averages sampled from the Q-function versus using the mean-action of the action-coordinate for computing the baseline (both described in 4.4). In our experiments, as shown in Figure 2 we find that the two variants perform comparably, with the latter performing slightly better towards the end of the learning process. This suggests that though sampling from the Q-function might provide a better estimate of the conditional expectation in theory, function approximation from finite samples injects errors that may degrade the quality of estimates. In particular, sub-sampling from the Q-function is likely to produce better results if the learned Q-function is accurate for a large fraction of the action space, but getting such high quality approximations might be hard in practice.

**High-dimensional action spaces**    Intuitively, the benefit of the action-dependent baseline can be greater for higher dimensional problems. We show this effect on a simple synthetic example called $m$-DimTargetMatching. The example is a one-step MDP comprising of a single state, $\mathcal{S} = \{0\}$, an $m$-dimensional action space, $\mathcal{A} = \mathbb{R}^m$, and a fixed vector $c \in \mathbb{R}^m$. The reward is given as the negative squared $\ell_2$ loss of the action vector, $r(s, a) = -\|a - c\|_2^2$. The optimal action is thus to match

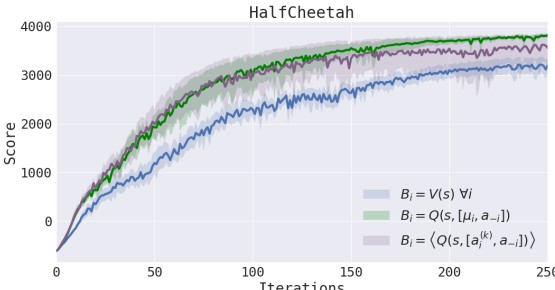

Figure 2: Variants of the action-dependent baseline that use: (i) sampling from the Q-function to estimate the conditional expectation; (ii) Using the mean action to form a linear approximation to the conditional expectation. We find that both variants perform comparably, with the latter being more computationally efficient.

the given vector by selecting $a = c$. The results for the demonstrative example are shown in Table 1, which shows that the action-dependent baseline successfully improves convergence more for higher dimensional problems than lower dimensional problems. Due to the lack of state information, the linear baseline reduces to whitening the returns. The action-dependent baseline, on the other hand, allows the learning algorithm to assess the advantage of each individual action dimension by utilizing information from all other action dimensions. Additionally, this experiment demonstrates that our algorithm scales well computationally to high-dimensional problems.

| Action | Solve time (iterations) | | | % speed | Solution |
|---|---|---|---|---|---|
| dimensions | Action-dependent | State-dependent | Delta | improvement | threshold |
| 12 | 45.6 | 45.6 | 0 | 0.0% | -0.01 |
| 100 | 136 | 150 | 14 | **9.3%** | -0.25 |
| 400 | 268.2 | 304 | 35.8 | **11.8%** | -0.99 |
| 2000 | 595.5 | 671.5 | 76 | **11.3%** | -4.96 |

Table 1: Shown are the results for the synthetic high-dimensional target matching task (5 seeds), for 12 to 2000 dimensional action spaces. At high dimensions, the linear feature action-dependent baseline provides notable and consistent variance reduction, as compared to a linear feature baseline, resulting in around 10% faster convergence. For the corresponding learning curves, see Appendix G.

**Partially observable and multi-agent tasks**   Finally, we also consider the extension of the core idea of using global information, by studying a POMDP task and a multi-agent task. We use the blind peg-insertion task which is widely studied in the robot learning literature (Montgomery & Levine, 2016). The task requires the robot to insert the peg into the hole (slot), but the robot is blind to the location of the hole. Thus, we expect a searching behavior to emerge from the robot, where it learns that the hole is present on the table and performs appropriate sweeping motions till it is able to find the hole. In this case, we consider a baseline that is given access to the location of the hole. We observe that a baseline with this additional information enables faster learning. For the multi-agent setting, we analyze a two-agent particle environment task in which the goal is for each agent to reach their goal, where their goal is known by the other agent and they have a continuous communication channel. Similar training procedures have been employed in recent related works Lowe et al. (2017); Levine et al. (2016). Figure 3 shows that including the inclusion of information from other agents into the action-dependent baseline improves the training performance, indicating that variance reduction may be key for multi-agent reinforcement learning.

## 6   CONCLUSION

An action-dependent baseline enables using additional signals beyond the state to achieve bias-free variance reduction. In this work, we consider both conditionally independent policies and general policies, and derive an optimal action-dependent baseline. We provide analysis of the variance

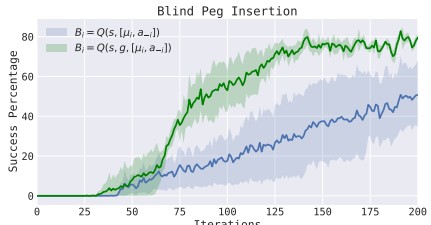
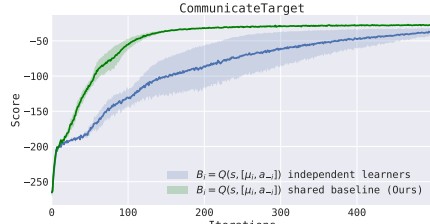

(a) Success percentage on the blind peg insertion task. The policy still acts on the observations and does not know the hole location. However, the baseline has access to this goal information, in addition to the observations and action, and helps to speed up the learning. By comparison, in blue, the baseline has access only to the observations and actions.

(b) Training curve for multi-agent communication task with two agents. Two policies are simultaneously trained, one for each agent. Each policy acts on the observations of its respective agent only. However, the shared baseline has access to the other agent's state and action, in addition to its own state and action, and results in considerably faster training. By comparison, in blue, the independent learners baseline has access to only a single agent's state and action.

Figure 3: Experiments with additional information in the baseline.

reduction improvement over non-optimal baselines, including the traditional optimal baseline that only depends on state. We additionally propose several practical action-dependent baselines which perform well on a variety of continuous control tasks and synthetic high-dimensional action problems. The use of additional signals beyond the local state generalizes to other problem settings, for instance in POMDP and multi-agent tasks. In future work, we propose to investigate related methods in such settings on large-scale problems.

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

## A    DERIVATION OF THE OPTIMAL STATE-DEPENDENT BASELINE

We provide a derivation of the optimal state-dependent baseline, which minimizes the variance of the policy gradient estimate, and is based in (Greensmith et al., 2004, Theorem 8). More precisely, we minimize the trace of the covariance of the policy gradient; that is, the sum of the variance of the components of the vectors. Recall the policy gradient expression with a state-dependent baseline:

$$\nabla_\theta \eta(\pi_\theta) := \mathbb{E}_{\rho_\pi,\pi} \left[ \nabla_\theta \log \pi_\theta(a_t|s_t) \left( \hat{Q}(s_t, a_t) - b(s_t) \right) \right] \tag{19}$$

Denote $g$ to be the associated random variable, that is, $\nabla_\theta \eta(\pi_\theta) = \mathbb{E}_{\rho_\pi,\pi}[g]$:

$$g := \nabla_\theta \log \pi_\theta(a_t|s_t) \left( \hat{Q}(s_t, a_t) - b(s_t) \right), \quad a_t \sim \pi_\theta(a_t|s_t), s_t \sim \rho_\pi(s_t) \tag{20}$$

The variance of the policy gradient is:

$$\text{Var}(g) = \mathbb{E}_{\rho_\pi,\pi} \left[ (g - \mathbb{E}_{\rho_\pi,\pi}[g])^T (g - \mathbb{E}_{\rho_\pi,\pi}[g]) \right] \tag{21}$$

$$= \mathbb{E}_{\rho_\pi,\pi} \left[ \nabla_\theta \log \pi_\theta(a_t|s_t)^T \nabla_\theta \log \pi_\theta(a_t|s_t) \right] b(s_t)^2 \tag{22}$$

$$- 2\mathbb{E}_{\rho_\pi,\pi} \left[ \nabla_\theta \log \pi_\theta(a_t|s_t)^T \nabla_\theta \log \pi_\theta(a_t|s_t) \hat{Q}(s_t, a_t) \right] b(s_t) \tag{23}$$

Note that $\mathbb{E}[\eta(\pi_\theta)])$ contains a bias-free term, by the score function argument, which then does not affect the minimizer. Terms which do not depend on $b(s_t)$ also do not affect the minimizer.

$$\frac{\partial}{\partial b} \left[ \text{Var}(g) \right] = 0 \tag{24}$$

$$= 2\mathbb{E}_{\rho_\pi,\pi} \left[ \nabla_\theta \log \pi_\theta(a_t|s_t)^T \nabla_\theta \log \pi_\theta(a_t|s_t) \right] b(s_t) \tag{25}$$

$$- 2\mathbb{E}_{\rho_\pi,\pi} \left[ \nabla_\theta \log \pi_\theta(a_t|s_t)^T \nabla_\theta \log \pi_\theta(a_t|s_t) \hat{Q}(s_t, a_t) \right] \tag{26}$$

$$\implies b^*(s_t) = \frac{\mathbb{E}_{\rho_\pi,\pi} \left[ \nabla_\theta \log \pi_\theta(a_t|s_t)^T \nabla_\theta \log \pi_\theta(a_t|s_t) \hat{Q}(s_t, a_t) \right]}{\mathbb{E}_{\rho_\pi,\pi} \left[ \nabla_\theta \log \pi_\theta(a_t|s_t)^T \nabla_\theta \log \pi_\theta(a_t|s_t) \right]} \tag{27}$$

## B    DERIVATION OF THE OPTIMAL ACTION-DEPENDENT BASELINE

We derive the optimal action-dependent baseline, which minimizes the variance of the policy gradient estimate. First, we write out the variance of the policy gradient under any action-dependent baseline. Recall the following notations: we define $z_i := \nabla_\theta \log \pi_\theta(a_t^i|s_t)$ and the component policy gradient:

$$\nabla \eta_i(\pi_\theta) := \mathbb{E}_{\rho_\pi,\pi} \left[ \nabla_\theta \log \pi_\theta(a_t^i|s_t) \left( \hat{Q}(s_t, a_t) - b_i(s_t, a_t^{-i}) \right) \right]. \tag{28}$$

Denote $g_i$ to be the associated random variables:

$$g_i := \nabla_\theta \log \pi_\theta(a_t^i|s_t) \left( \hat{Q}(s_t, a_t) - b_i(s_t, a_t^{-i}) \right), \quad a_t \sim \pi_\theta(a_t|s_t), s_t \sim \rho_\pi(s_t), \tag{29}$$

such that

$$\nabla_\theta \eta(\pi_\theta) = \nabla_\theta \left[ \sum_{i=1}^m \eta_i(\pi_\theta) \right] = \mathbb{E}_{\rho_\pi,\pi} \left[ \sum_{i=1}^m g_i \right]. \tag{30}$$

Recall the following assumption:

$$\nabla_\theta \log \pi_\theta(a_t^i|s_t)^T \nabla_\theta \log \pi_\theta(a_t^j|s_t) \equiv z_i^T z_j \approx 0, \quad \forall i \neq j, \tag{31}$$

which translates to meaning that different subsets of parameters strongly influence different action dimensions or factors. This is true in case of distributed systems by construction, and also true

in a single agent system if different action coordinates are strongly influenced by different policy network channels. Under this assumption, we have:

$$\text{Var}(\sum_{i=1}^{m} g_i) = \sum_i \text{Var}(g_i) + \sum_i \sum_{j \neq i} \text{Cov}(g_i, g_j) \tag{32}$$

$$= \sum_i \text{Var}(g_i) + \sum_i \sum_{j \neq i} \mathbb{E}_{\rho_\pi, \pi} \left[ g_i^T g_j \right] - \mathbb{E}_{\rho_\pi, \pi} \left[ g_i \right]^T \mathbb{E}_{\rho_\pi, \pi} \left[ g_j \right] \tag{33}$$

$$= \sum_i \text{Var}(g_i) + 0 - \sum_i \sum_{j \neq i} \mathbb{E}_{\rho_\pi, \pi} \left[ g_i \right]^T \mathbb{E}_{\rho_\pi, \pi} \left[ g_j \right] \quad \text{(by Equation (31))} \tag{34}$$

$$= \sum_i \text{Var}(g_i) - \sum_i \sum_{j \neq i} M_{ij} \quad \text{(by score function estimator)} \tag{35}$$

where we denote the mean correction term $M_{ij} := \mathbb{E}_{\rho_\pi, \pi} \left[ z_i \hat{Q}(s_t, a_t) \right]^T \mathbb{E}_{\rho_\pi, \pi} \left[ z_j \hat{Q}(s_t, a_t) \right]$. Also let $M = \sum_i \sum_j M_{ij}$. Note that $M$ does not depend on $b_i(\cdot)$, and thus does not affect the optimal value.

The overall variance is minimized when each component variance is minimized. We now derive the optimal baselines $b_i^*(s_t, a_t^{-i})$ which minimize each respective component.

$$\text{Var}(g_i) = \mathbb{E}_{\rho_\pi, \pi} \left[ z_i^T z_i \left( \hat{Q}(s_t, a_t) - b_i(s_t, a_t^{-i}) \right)^2 \right] \tag{36}$$

$$- \mathbb{E}_{\rho_\pi, \pi} \left[ z_i \left( \hat{Q}(s_t, a_t) - b_i(s_t, a_t^{-i}) \right) \right]^T \mathbb{E}_{\rho_\pi, \pi} \left[ z_i \left( \hat{Q}(s_t, a_t) - b_i(s_t, a_t^{-i}) \right) \right]$$

$$= \mathbb{E}_{\rho_\pi, \pi} \left[ z_i^T z_i \left( \hat{Q}(s_t, a_t)^2 - 2b_i(s_t, a_t^{-i}) \hat{Q}(s_t, a_t) + b_i(s_t, a_t^{-i})^2 \right) \right] \tag{37}$$

$$- \mathbb{E}_{\rho_\pi, \pi} \left[ z_i \left( \hat{Q}(s_t, a_t) \right) \right]^T \mathbb{E}_{\rho_\pi, \pi} \left[ z_i \left( \hat{Q}(s_t, a_t) \right) \right]$$

$$= \mathbb{E}_{\rho_\pi, \pi} \left[ z_i^T z_i \hat{Q}(s_t, a_t)^2 \right] \tag{38}$$

$$+ \mathbb{E}_{\rho_\pi, a_t^{-i}} \left[ -2b_i(s_t, a_t^{-i}) \mathbb{E}_{a_t^i} \left[ z_i^T z_i \hat{Q}(s_t, a_t) \right] + b_i(s_t, a_t^{-i})^2 \mathbb{E}_{a_t^i} \left[ z_i^T z_i \right] \right] - M_{ii}$$

Having written down the expression for variance under any action-dependent baseline, we seek the optimal baseline that would minimize this variance.

$$\frac{\partial}{\partial b_i} \left[ \text{Var}(\sum_i g_i) \right] = \frac{\partial}{\partial b_i} \left[ \text{Var}(g_i) \right] = 0 \tag{39}$$

$$\implies b_i^*(s_t, a_t^{-i}) = \frac{\mathbb{E}_{a_t^i} \left[ z_i^T z_i \hat{Q}(s_t, a_t) \right]}{\mathbb{E}_{a_t^i} \left[ z_i^T z_i \right]} \tag{40}$$

The optimal action-dependent baseline is:

$$b_i^*(s_t, a_t^{-i}) = \frac{\mathbb{E}_{a_t^i} \left[ \nabla_\theta \log \pi_\theta(a_t^i | s_t)^T \nabla_\theta \log \pi_\theta(a_t^i | s_t) \hat{Q}(s_t, a_t) \right]}{\mathbb{E}_{a_t^i} \left[ \nabla_\theta \log \pi_\theta(a_t^i | s_t)^T \nabla_\theta \log \pi_\theta(a_t^i | s_t) \right]} \tag{41}$$

## C   DERIVATION OF VARIANCE REDUCTION IMPROVEMENT

We now turn to quantifying the reduction in variance of the policy gradient estimate under the optimal baseline derived above. Let $\text{Var}^*(\sum_i g_i)$ denote the variance resulting from the optimal action-dependent baseline, and let $\text{Var}(\sum_i g_i)$ denote the variance resulting from another baseline

$b = (b_i(s_t, a_t^{-i}))_{i \in [m]}$, which may be suboptimal or action-independent. Recall the notation:

$$Z_i := Z_i(s_t, a_t^{-i}) = \mathbb{E}_{a_t^i} \left[ \nabla_\theta \log \pi_\theta(a_t^i|s_t)^T \nabla_\theta \log \pi_\theta(a_t^i|s_t) \right] \tag{42}$$

$$Y_i := Y_i(s_t, a_t^{-i}) = \mathbb{E}_{a_t^i} \left[ \nabla_\theta \log \pi_\theta(a_t^i|s_t)^T \nabla_\theta \log \pi_\theta(a_t^i|s_t) \hat{Q}(s_t, a_t) \right] \tag{43}$$

$$X_i := X_i(s_t, a_t^{-i}) = \mathbb{E}_{a_t^i} \left[ \nabla_\theta \log \pi_\theta(a_t^i|s_t)^T \nabla_\theta \log \pi_\theta(a_t^i|s_t) \hat{Q}(s_t, a_t)^2 \right] \tag{44}$$

Finally, define the variance improvement $I_b := \mathrm{Var}(\sum_i g_i) - \mathrm{Var}^*(\sum_i g_i)$. Using these definitions, the variance can be re-written as:

$$\mathrm{Var}(\sum_i g_i) = \sum_i \mathbb{E}_{\rho_\pi, a_t^{-i}} \left[ X_i - 2b_i(s_t, a_t^{-i})Y_i + b_i(s_t, a_t^{-i})^2 Z_i \right] - M \tag{45}$$

Furthermore, the variance of the gradient with the optimal baseline can be written as

$$\mathrm{Var}^*(\sum_i g_i) = \sum_i \mathbb{E}_{\rho_\pi, a_t^{-i}} \left[ X_i - \frac{Y_i^2}{Z_i} \right] - M \tag{46}$$

The difference in variance can be calculated as:

$$I_b := \sum_i \left( \mathbb{E}_{\rho_\pi, a_t^{-i}} \left[ X_i - 2b_i(s_t, a_t^{-i})Y_i + b_i(s_t, a_t^{-i})^2 Z_i \right] - \left( \mathbb{E}_{\rho_\pi, a_t^{-i}} \left[ X_i - \frac{Y_i^2}{Z_i} \right] \right) \right) \tag{47}$$

$$= \sum_i \mathbb{E}_{\rho_\pi, a_t^{-i}} \left[ -2b_i(s_t, a_t^{-i})Y_i + b_i(s_t, a_t^{-i})^2 Z_i + \frac{Y_i^2}{Z_i} \right] \tag{48}$$

$$= \sum_i \mathbb{E}_{\rho_\pi, a_t^{-i}} \left[ \left( b_i(s_t, a_t^{-i})\sqrt{Z_i} - \frac{Y_i}{\sqrt{Z_i}} \right)^2 \right] \tag{49}$$

$$= \sum_i \mathbb{E}_{\rho_\pi, a_t^{-i}} \left[ Z_i \left( b_i(s_t, a_t^{-i}) - \frac{Y_i}{Z_i} \right)^2 \right] \tag{50}$$

$$= \sum_i \mathbb{E}_{\rho_\pi, a_t^{-i}} \left[ Z_i \left( b_i(s_t, a_t^{-i}) - b_i^*(s_t, a_t^{-i}) \right)^2 \right] \tag{51}$$

$$= \sum_i \mathbb{E}_{\rho_\pi, a_t^{-i}} \left[ \mathbb{E}_{a_t^i} \left[ \nabla_\theta \log \pi_\theta(a_t^i|s_t)^T \nabla_\theta \log \pi_\theta(a_t^i|s_t) \right] \left( b_i(s_t, a_t^{-i}) - b_i^*(s_t, a_t^{-i}) \right)^2 \right] \tag{52}$$

## D  DERIVATION OF SUBOPTIMALITY OF THE OPTIMAL STATE-DEPENDENT BASELINE

Using the notation from Appendix C and working off of Equation (51), we have:

$$I_{b=b^*(s)} := \sum_i \mathbb{E}_{\rho_\pi, a_t^{-i}} \left[ Z_i \left( b^*(s_t) - b_i^*(s_t, a_t^{-i}) \right)^2 \right] \tag{53}$$

$$= \sum_i \mathbb{E}_{\rho_\pi, a_t^{-i}} \left[ Z_i \left( \frac{\sum_j Y_j}{\sum_j Z_j} - \frac{Y_i}{Z_i} \right)^2 \right] \tag{54}$$

$$= \sum_i \mathbb{E}_{\rho_\pi, a_t^{-i}} \left[ \frac{1}{Z_i} \left( \frac{Z_i}{\sum_j Z_j} \sum_j Y_j - Y_i \right)^2 \right] \tag{55}$$

## E  BASELINES FOR GENERAL ACTIONS

In the preceding derivations, we have assumed policy actions are conditionally independent across dimensions. In the more general case, we only assume that there are $m$ factors $a_t^1$ through $a_t^m$

which altogether forms the action $a_t$. Conditioned on $s_t$, the different factors form a certain directed acyclic graphical model (including the fully dependent case). Without loss of generality, we assume that the following factorization holds:

$$\pi_\theta(a_t|s_t) = \prod_{i=1}^{m} \pi_\theta(a_t^i|s_t, a_t^{f(i)}) \tag{56}$$

where $f(i)$ denotes the indices of the parents of the $i$th factor. Let $D(i)$ denote the indices of descendants of $i$ in the graphical model (including $i$ itself). In this case, we can set the $i$th baseline to be $b_i(s_t, a_t^{[m]\backslash D(i)})$, where $[m] = \{1, 2, \ldots, m\}$. In other words, the $i$th baseline can depend on all other factors which the $i$th factor does not influence. The overall gradient estimator is given by

$$\nabla_\theta \eta(\pi_\theta) = \mathbb{E}_{\rho_\pi, \pi}\left[\sum_{i=1}^{m} \nabla_\theta \log \pi_\theta(a_t^i|s_t, a_t^{f(i)})\left(\hat{Q}(s_t, a_t) - b_i(s_t, a_t^{[m]\backslash D(i)})\right)\right] \tag{57}$$

In the most general case without any conditional independence assumptions, we have $f(i) = \{1, 2, \ldots, i-1\}$, and $D(i) = \{i, i+1, \ldots, m\}$. The above equation reduces to

$$\nabla_\theta \eta(\pi_\theta) = \mathbb{E}_{\rho_\pi, \pi}\left[\sum_{i=1}^{m} \nabla_\theta \log \pi_\theta(a_t^i|s_t, a_t^1, \ldots, a_t^{i-1})\left(\hat{Q}(s_t, a_t) - b_i(s_t, a_t^1, \ldots, a_t^{i-1})\right)\right] \tag{58}$$

The above analysis for optimal baselines and variance suboptimality transfers also to the case of general actions.

The applicability of our techniques to general action spaces may be of crucial importance for many application domains where the conditional independence assumption does not hold up, such as language tasks and other compositional domains. Even in continuous control tasks, such as hand manipulation, and many other tasks where it is common practice to use conditionally independent factorized policies, it is reasonable to expect training improvement from policies without a full conditionally independence structure.

**Computing action-dependent baselines for general actions** The marginalization presented in Section 4.4 does not apply for the general action setting. Instead, $m$ individual baselines can be trained according to the factorization, and each of them can be fitted from data collected from the previous iteration. In the general case, this means fitting $m$ functions $b_i(s_t, a_t^1, \ldots, a_t^{i-1})$, for $i \in \{1, \ldots, m\}$. The resulting method is described in Algorithm 2.

There may also exist special cases like conditional independent actions, for which more efficient baseline constructions exist. A closely related example to the conditionally independent case is the case of block diagonal covariance structures (e.g. in multi-agent settings), where we may wish to instead learn an overall Q function and marginalize over block factors. Another interesting example to explore is sparse covariance structures.

---

**Algorithm 2** Policy gradient for general factorization policies using action-dependent baselines

---

**Require:** number of iterations $N$, batch size $B$, initial policy parameters $\theta$
    Initialize baselines $b_i(s_t, a_t^{[m]\backslash D(i)}) \equiv 0$, for $i \in \{1, \ldots, m\}$ and policy $\pi_\theta$
    **for** $j$ in $\{1, \ldots, N\}$ **do**
        Collect samples: $(s_t, a_t)_{t \in \{1, \ldots, B\}}$
        Compute advantages: $\hat{A}_i(s_t, a_t) := \hat{Q}(s_t, a_t) - b_i(s_t, a_t^{[m]\backslash D(i)}), \forall t$
        Perform a policy update step on $\theta$ using $\hat{A}_i(s_t, a_t)$ [Equation (57)]
        Update baseline functions with current batch: $b_i(s_t, a_t^{[m]\backslash D(i)})$
    **end for**

---

## F  COMPATIBILITY WITH GAE

Temporal Difference (TD) learning methods such as GAE (Schulman et al., 2016) allow us to smoothly interpolate between high-bias, low-variance estimates and low-bias, high-variance estimates of the policy gradient. These methods are based on the idea of being able to predict future

returns, thereby bootstrapping the learning procedure. In particular, when using the value function as a baseline, we have $A(s_t, a_t) = \mathbb{E}\left[r_t + \gamma V(s_{t+1}) - V(s_t)\right] = \left[r_t + \gamma b(s_{t+1}) - b(s_t)\right]$ if $b(s)$ is an unbiased estimator for $V(s)$. GAE uses an exponential averaging of such temporal difference terms over a trajectory to significantly reduce the variance of the advantage at the cost of a small bias (it allows us to pick where we want to be on the bias-variance curve). Similarly, if we use $b_i(s_t, a_t^{-i})$ as an unbiased estimator for $\mathbb{E}_{a_t^i}[\hat{Q}(s_t, a_t)]$, we have:

$$\mathbb{E}_{\pi, \mathcal{M}}\left[r_t + \gamma b_i(s_{t+1}, a_{t+1}^{-i}) - b_i(s_t, a_t^{-i})\right] = Q(s_t, a_t) - \mathbb{E}_{a_t^i}[\hat{Q}(s_t, a_t)] = A_i(s_t, a_t) \quad (59)$$

Thus, the temporal difference error with the action dependent baselines is an unbiased estimator for the advantage function as well. This allows us to use the GAE procedure to further reduce variance at the cost of some bias.

The following study shows that action-dependent baselines are consistent with TD procedures with their temporal differences being estimates of the advantage function. Our results summarized in Figure 4 suggests that slightly biasing the gradient to reduce variance produces the best results, while high-bias estimates perform poorly. Prior work with baselines that utilize global information (Foerster et al., 2017) employ the high-bias variant. The results here suggest that there is potential to further improve upon those results by carefully studying the bias-variance trade-off.

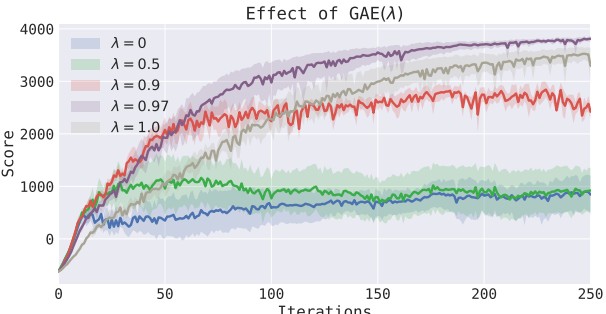

Figure 4: We study the influence of $\lambda$ in GAE which allows us to trade off bias and variance as desired. High bias gradient corresponding to smaller values of $\lambda$ do not make progress after a while. High variance gradient ($\lambda = 1$) has trouble learning initially. Allowing for a small bias to reduce the variance, corresponding to the intermediate $\lambda = 0.97$ produces the best overall result, consistent with the findings in Schulman et al. (2016).

## G  HIGH-DIMENSIONAL ACTION SPACES: TRAINING CURVES

Figure 5 shows the resulting training curves for a synthetic high-dimensional target matching task, as described in Section 5. For higher dimensional action spaces (100 dimensions or greater), the action-dependent baseline consistently converges to the optimal solution 10% faster than the state-only baseline.

For reference, Figure 6 shows the result of the original high-dimensional action space experiment. Due to a discovered issue in the TensorFlow version of rllab, which results in training instability, both methods (action-dependent and state-dependent baselines) under-performed relative to the revised experiment (Figure 5), which uses a clean implementation based on the implementation referenced inRajeswaran et al. (2017b). The regression in training is most evident by the number of iterations required to solve the task; for instance, the old experiment could take as long as five times more iterations to solve the same task, even for a 12-dimensional task.

## H  EXPERIMENT DETAILS

**Parameters**: Unless otherwise stated, the following parameters are used in the experiments in this work: $\gamma = 0.995$, $\lambda_{\text{GAE}} = 0.97$, $kl_{\text{desired}} = 0.025$.

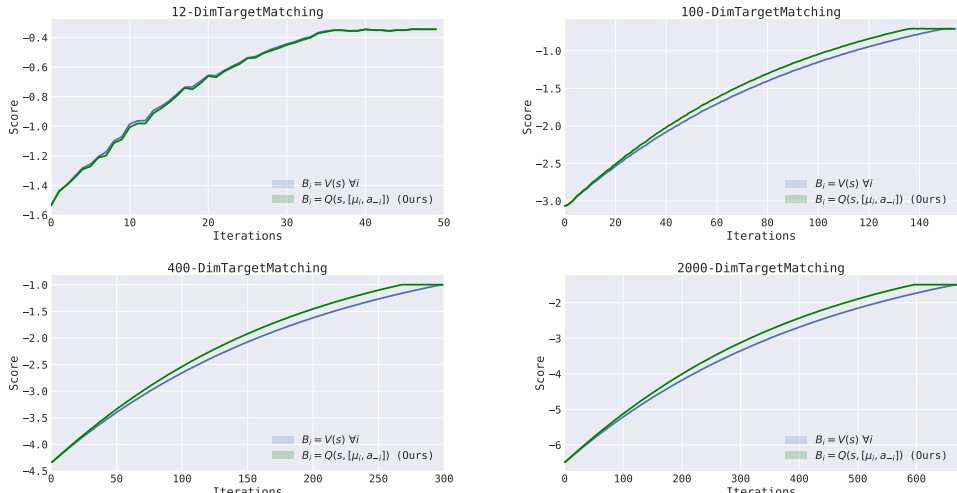

Figure 5: Shown is the learning curve for a synthetic high-dimensional target matching task (5 seeds), for 12 to 2000 dimensional action spaces. At high dimensions, the linear feature action-dependent baseline provides notable and consistent variance reduction, as compared to a linear feature state baseline. For 100, 400, and 2000 dimensions, our method converges 10% faster to the optimal solution.

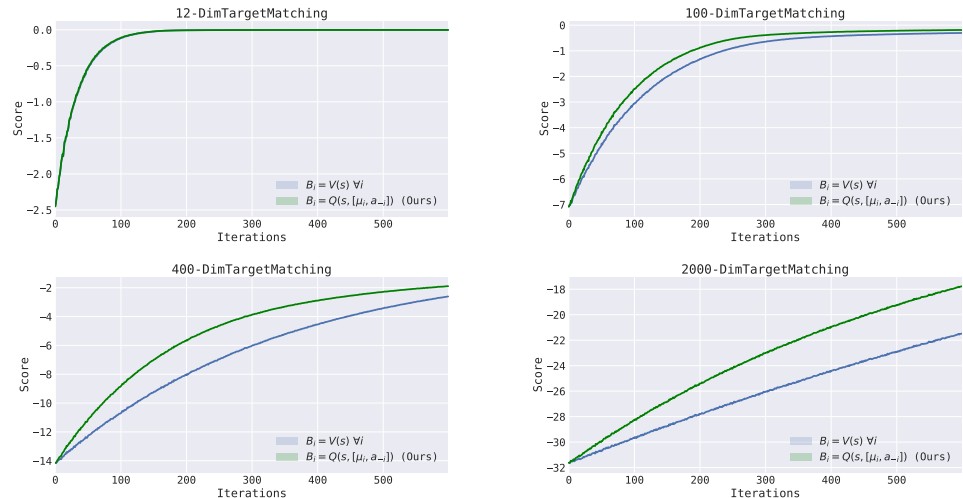

Figure 6: Reference training curves for an early and invalidated version of the synthetic high-dimensional target matching task (3 seeds).

**Policies**: The policies used are 2-layer fully connected networks with hidden sizes=(32, 32).

**Initialization**: the policy is initialized with Xavier initialization except final layer weights are scaled down (by a factor of 100x). Note that since the baseline is linear (with RBF features) and estimated with a Newton step, the initialization is inconsequential.

**Per-experiment configuration**: The following parameters in Table 2 are for both state-only and action-dependent versions of the experiments. The $m$-DimTargetMatching experiments use a linear feature baseline. Table 3 details the dimensionality of the action space for each task.

| Task | Benchmarks | Hand task | Peg Insertion | CommunicateTarget | $m$-DimTargetMatching |
|---|---|---|---|---|---|
| Trajectories | 10 | 100 | 200 | 300 | 150 |
| Horizon | 1000 | 200 | 250 | 100 | 1 |
| RBF features | 100 | 250 | 250 | 250 | N/A |

Table 2: Experiment details

| Task | Action dimensions |
|---|---|
| Hopper | 3 |
| HalfCheetah | 6 |
| Ant | 8 |
| Hand | 30 |
| Peg | 7 |
| CommunicateTarget | 8 (4 per agent) |
| $m$-DimTargetMatching | $m$ |

Table 3: Action dimensionality of tasks

