# OpenReview forum: "Variance Reduction for Policy Gradient with Action-Dependent Factorized Baselines"
_ICLR.cc/2018/Conference — Accept (Oral)_

### Official Review · AnonReviewer2 · 2017-11-26
**This paper presents methods to reduce the variance of policy gradient using action dependent baselines when actions have conditionally independent factors.**

**Rating:** 7
**Confidence:** 4

**Review:**

This paper presents methods to reduce the variance of policy gradient using an action dependent baseline. Such action dependent baseline can be used in settings where the action can be decomposed into factors that are conditionally dependent given the state. The paper:
(1) shows that using separate baselines for actions, each of which can depend on the state and other actions is bias-free
(2) derive the optimal action-dependent baseline, showing that it does not degenerate into state-only dependent baseline, i.e. there is potentially room for improvement over state-only baselines.
(3) suggests using marginalized action-value (Q) function as a practical baseline, generalizing the use of value function in state-only baseline case.
(4) suggests using MC marginalization and also using the "average" action to improve computational feasibility
(5) combines the method with GAE techniques to further improve convergence by trading off bias and variance

The suggested methods are empirically evaluated on a number of settings. Overall action-dependent baseline outperform state-only versions. Using a single average action marginalization is on par with MC sampling, which the authors attribute to the low quality of the Q estimate. Combining GAE shows that a hint of bias can be traded off with further variance reduction to further improve the performance.

I find the paper interesting and practical to the application of policy gradient in high dimensional action spaces with some level of conditional independence present in the action space. In light of such results, one might change the policy space to enforce such structure.

Notes:
- Elaborate further on the assumption made in Eqn 9. Does it mean that the actions factors cannot share (too many) parameters in the policy construction, or that shared parameters can only be applied to the state?
- Eqn 11 should use \simeq
- How can the notion of average be extended to handle multi-modal distributions, or categorical or structural actions? Consider expanding on that in section 4.5.
- The discussion on the DAG graphical model is lacking experimental analysis (where separate baselines models are needed). How would you train such baselines?
- Figure 4 is impossible to read in print. The fonts are too small for the numbers and the legends.

---

> ### Author Response · Authors · 2018-01-04
> **Thank you**
>
> Thank you for the thorough review! We have updated the paper based on your suggestions. We have added discussions on categorical distributions to Section 4.4, sub-section 2 and discussions on the generic DAG graphical model to Appendix E, last 3 paragraphs. We have addressed your key point below and incorporated the discussion into the revised article.
>
> > Elaborate further on the assumption made in Eqn 9. Does it mean that the actions
> > factors cannot share (too many) parameters in the policy construction, or that shared
> > parameters can only be applied to the state?
>
> The assumption made in Eqn 9 is primarily for the theoretical analysis to be clean, and is not required to run the algorithm in practice. In particular, even without this assumption, the proposed baseline is bias-free. When the assumption holds, the optimal action-dependent baseline has a clean form which we can analyze thoroughly. As noted by the reviewer, the assumption is not very unrealistic. Some examples where these assumptions hold include multi-agent settings where the policies are conditionally independent by construction, cases where the policy acts based on independent components [1] of the observation space, and cases where different function approximators are used to control different actions or synergies [2,3] without weight sharing.
>
> [1] Y. Cao et al. Motion Editing With Independent Component Analysis, 2007.
> [2] E. Todorov, Z. Ghahramani, Analysis of the synergies underlying complex hand manipulation, 2004.
> [3] E. Todorov, W. Li, X. Pan, From task parameters to motor synergies: A hierarchical framework for approximately optimal control of redundant manipulators, 2005.

---

### Official Review · AnonReviewer3 · 2017-11-27

**Rating:** 8
**Confidence:** 3

**Review:**

In this paper, the authors investigate variance reduction techniques for agents with multi-dimensional policy outputs, in particular when they are conditionally independent ('factored'). With the increasing focus on applying RL methods to continuous control problems and RTS type games, this is an important problem and this technique seems like an important addition to the RL toolbox. The paper is well written, the method is easy to implement, and the algorithm seems to have clear positive impact on the presented experiments.

- The derivations in pages 4-6 are somewhat disconnected from the rest of the paper: the optimal baseline derivation is very standard (even if adapted to the slightly different situation situated here), and for reasons highlighted by the authors in this paper, they are not often used; the 'marginalized' baseline is more common, and indeed, the authors adopt this one as well. In light of this (and of the paper being quite a bit over the page limit)- is this material (4.2->4.4) mostly not better suited for the appendix? Same for section 4.6 (which I believe is not used in the experiments).

- The experimental section is very strong; regarding the partial observability experiments, assuming actions are here factored as well, I could see four baselines
(two choices for whether the baseline has access to the goal location or not, and two choices for whether the baseline has access to the vector $a_{-i}$). It's not clear which two baselines are depicted in 5b - is it possible to disentangle the effect of providing $a_{-i}$ and the location of the hole to the baseline?

(side note: it is an interesting idea to include information not available to the agent as input to the baseline though it does feel a bit 'iffy' ; the agent requires information to train, but is not provided the information to act.  Out of curiosity, is it intended as an experiment to verify the need for better baselines? Or as a 'fair' training procedure?)

- Minor: in equation 2- is the correct exponent not t'?  Also since $\rho_\pi$ is define with a scaling $(1-\gamma)$ (to make it an actual distribution), I believe the definition of $\eta$ should also be multiplied by $(1-\gamma)$ (as well as equation 2).

---

> ### Author Response · Authors · 2018-01-04
> **Thank you**
>
> Thank you for the clear and encouraging review! We have addressed your key points below and incorporated the discussion into the revised article.
>
> > The derivations in pages 4-6 are somewhat disconnected from the rest of the paper: the
> > optimal baseline derivation is very standard (even if adapted to the slightly different
> > situation situated here), and for reasons highlighted by the authors in this paper, they
> > are not often used; the 'marginalized' baseline is more common, and indeed, the authors
> > adopt this one as well. In light of this (and of the paper being quite a bit over the page
> > limit)- is this material (4.2->4.4) mostly not better suited for the appendix? Same for
> > section 4.6 (which I believe is not used in the experiments).
>
> Thank you for your suggestion. We have moved the derivation and the general actions exposition to Appendices B-D and E, respectively, and have referenced only the important conclusions in the main text.
>
> > The experimental section is very strong; regarding the partial observability experiments,
> > assuming actions are here factored as well, I could see four baselines (two choices for
> > whether the baseline has access to the goal location or not, and two choices for whether
> > the baseline has access to the vector $a_{-i}$). It's not clear which two baselines are
> > depicted in 5b - is it possible to disentangle the effect of providing $a_{-i}$ and the
> > location of the hole to the baseline?
> >
> > (side note: it is an interesting idea to include information not available to the agent as
> > input to the baseline though it does feel a bit 'iffy' ; the agent requires information to
> > train, but is not provided the information to act. Out of curiosity, is it intended as an
> > experiment to verify the need for better baselines? Or as a 'fair' training procedure?)
>
> Thank you for this observation. We have updated the experiments to compare baseline1=state+action+goal vs baseline2=state+action, and have generated results for more random seeds (5). Similarly, the multi-agent experiment is comparing whether the baseline has access to the state of other agents or not, in addition to a single agent’s state+action. Our primary goal in these experiments was to see if providing additional information can reduce variance and help train faster. At test time, both policies are required to act based on the same information, and hence this is a ‘fair’ procedure. Similar approaches of using additional information during training time have been employed in recent related works [1,2], which we have referenced in the paper.
>
> [1] Lowe et al. Multi-Agent Actor-Critic for Mixed Cooperative-Competitive Environments, 2017.
> [2] Levine, et al. End-to-end training of deep visuomotor policies, 2016.
>
> > Minor: in equation 2- is the correct exponent not t'? Also since $\rho_\pi$ is define with
> > a scaling $(1-\gamma)$ (to make it an actual distribution), I believe the definition of
> > $\eta$ should also be multiplied by $(1-\gamma)$ (as well as equation 2).
>
> Thank you for the detailed questions and comments! The correct exponent is $t’-t$ because what is being computed is the cumulative discounted return starting from time t. Thank you also for catching our error with the $(1-\gamma)$. We have corrected this in the manuscript in Section 3.3.

---

### Official Review · AnonReviewer1 · 2017-11-27
**Useful idea for variance reduction with some issues in the experiments**

**Rating:** 6
**Confidence:** 4

**Review:**

The paper proposes a variance reduction technique for policy gradient methods. The proposed approach justifies the utilization of action-dependent baselines, and quantifies the gains achieved by it over more general state-dependent or static baselines.


The writing and organization of the paper is very well done. It is easy to follow, and succinct while being comprehensive. The baseline definition is well-motivated, and the benefits offered by it are quantified intuitively. There is only one mostly minor issues with the algorithm development and the experiments need to be more polished.

For the algorithm development, there is an relatively strong assumption that z_i^T z_j = 0. This assumption is not completely unrealistic (for example, it is satisfied if completely separate parts of a feature vector are used for actions). However, it should be highlighted as an assumption, and it should be explicitly stated as z_i^T z_j = 0 rather than z_i^T z_j approx 0. Further, because it is relatively strong of an assumption, it should be discussed more thoroughly, with some explicit examples of when it is satisfied.

Otherwise, the idea is simple and yet effective, which is exactly what we would like for our algorithms. The paper would be a much stronger contribution, if the experiments could be improved.
- More details regarding the experiments are desirable - how many runs were done, the initialization of the policy network and action-value function, the deep architecture used etc.
- The experiment in Figure 3 seems to reinforce the influence of \lambda as concluded by the Schulman et. al. paper. While that is interesting, it seems unnecessary/non-relevant here, unless performance with action-dependent baselines with each value of \lambda is contrasted to the state-dependent baseline. What was the goal here?
- In general, the graphs are difficult to read; fonts should be improved and the graphs polished.
- The multi-agent task needs to be explained better - specifically how is the information from the other agent incorporated in an agent's baseline?
- It'd be great if Plot (a) and (b) in Figure 5 are swapped.

Overall I think the idea proposed in the paper is beneficial. Better discussing the strong theoretical assumption should be incorporated. Adding the listed suggestions to the experiments section would really help highlight the advantage of the proposed baseline in a more clear manner. Particularly with some clarity on the experiments, I would be willing to increase the score.

Minor comments:
1. In Equation (28) how is the optimal-state dependent baseline obtained? This should be explicitly shown, at least in the appendix.
2. The listed site for videos and additional results is not active.
3. Some typos
- Section 2 - 1st para - last line: "These methods are therefore usually more sample efficient, but can be less stable than critic-based methods.".
- Section 4.1 - Equation (7) - missing subscript i for b(s_t,a_t^{-i})
- Section 4.2 - \hat{Q} is just Q in many places

---

> ### Author Response · Authors · 2018-01-04
> **Thank you**
>
> Thank you for the thoughtful review! These suggestions and questions are reflected in the updated article. We have added the derivation of the optimal-state dependent baseline (based on Greensmith, et al., 2004) in Appendix A, and a video has since been uploaded to the site. We have added experiment details to Appendix G, and have clarified baselines for multi-agent settings in Section 5 (Figure 4b). Thank you also for noting the typos! We have updated the manuscript to reflect these changes and included a clarification of our notation in Section 3.1, paragraph 1. We have addressed your key points below and incorporated the discussion into the new revision of the article.
>
> > For the algorithm development, there is an relatively strong assumption that z_i^T z_j = 0. This
> > assumption is not completely unrealistic (for example, it is satisfied if completely separate parts
> > of a feature vector are used for actions). However, it should be highlighted as an assumption,
> > and it should be explicitly stated as z_i^T z_j = 0 rather than z_i^T z_j approx 0. Further, because
> > it is relatively strong of an assumption, it should be discussed more thoroughly, with some
> > explicit examples of when it is satisfied.
>
> Thank you for this very important observation. We have revised the manuscript to state this assumption explicitly and have also provided examples where it is satisfied (in Section 4.2, paragraph 1). We note however that this assumption is primarily for the theoretical analysis to be clean, and is not required to run the algorithm in practice. In particular, even without this assumption, the proposed baseline is bias-free. When the assumption holds, the optimal action-dependent baseline has a clean form which we can analyze thoroughly. As noted by the reviewer, the assumption is not very realistic. Some examples where these assumptions hold include multi-agent settings where the policies are conditionally independent by construction, cases where the policy acts based on independent components [1] of the observation space, and cases where different function approximators are used to control different actions or synergies [2,3] without weight sharing.
>
> [1] Y. Cao et al. Motion Editing With Independent Component Analysis, 2007.
> [2] E. Todorov, Z. Ghahramani, Analysis of the synergies underlying complex hand manipulation, 2004.
> [3] E. Todorov, W. Li, X. Pan, From task parameters to motor synergies: A hierarchical framework for approximately optimal control of redundant manipulators, 2005.
>
> > The experiment in Figure 3 seems to reinforce the influence of \lambda as concluded by the
> > Schulman et. al. paper. While that is interesting, it seems unnecessary/non-relevant here,
> > unless performance with action-dependent baselines with each value of \lambda is contrasted
> > to the state-dependent baseline. What was the goal here?
>
> Thank you for the great question. Our goal was to emphasize that one does not lose out on temporal difference based variance reduction approaches like GAE, which are complimentary to reducing variance caused by high dimensionality of action space considered in this work. Considering the page limit and your suggestion, we have moved this discussion to Appendix F.

---

### Author Response · Authors · 2018-01-04
**Thank you**

We thank reviewers for their time and thoughtful feedback.

We have updated the submission: We have moved the derivations and extensions to the appendix, added a summarizing algorithms section. We have improved notation throughout the paper, improved the consistency of the plots, clarified experiment details, and resolved ambiguities. We answer specific questions raised in the reviews by separately replying to each of them.

---

### Public Comment · (anonymous) · 2018-02-07
**The derivations presented in the Appendix appear to have correctness or clarity issues**

Thank you for your work!

Some derivations in the paper appear to have issues.

For instance, Equation 15 (Appendix A) presents the gradient of the expected return, which is a constant for a fixed theta. Equation 16 proceeds to present the variance of this constant. In all likelihood, the authors meant to present the variance of the estimator (random vector inside the expectation in the right hand size of Equation 15). This apparent mistake is also present in other sections of the paper.

Equation 19 also appears problematic, since the baseline is a function of the state, a fact that does not seem to be taken into account in Equation 17. I believe this step deserves clarification.

---

> ### Author Response · Authors · 2018-02-25
> **Thank you for the comment!**
>
> Thank you for your constructive comments! We have updated Appendix A to clearly handle random variables representing the policy gradient instead of its expected value.
>
> The updated version can be found here: https://openreview.net/forum?id=HkuBsbb0b

---

### Decision · Program_Chairs · 2018-01-29
**ICLR 2018 Conference Acceptance Decision**

**Decision:**

Accept (Oral)

**Comment:**

The reviewers are satisfied that this paper makes a good contribution to policy gradient methods.